# Potential Risk of Three Zoonotic Protozoa (*Cryptosporidium* spp., *Giardia duodenalis*, and *Toxoplasma gondii*) Transmission from Fish Consumption

**DOI:** 10.3390/foods9121913

**Published:** 2020-12-21

**Authors:** Samantha Moratal, M. Auxiliadora Dea-Ayuela, Jesús Cardells, Naima M. Marco-Hirs, Silvia Puigcercós, Víctor Lizana, Jordi López-Ramon

**Affiliations:** 1Servicio de Análisis, Investigación y Gestión de Animales Silvestres (SAIGAS), Veterinary Faculty, Universidad CEU-Cardenal Herrera, Tirant lo Blanc St 7, 46115 Alfara del Patriarca, Valencia, Spain; samantha.moratalmartinez@uchceu.es (S.M.); jcardells@uchceu.es (J.C.); naima.marco@uchceu.es (N.M.M.-H.); silvia.puigcercosgomila@uchceu.es (S.P.); victor.lizana@uchceu.es (V.L.); jordi.lopez1@uchceu.es (J.L.-R.); 2Farmacy Department, Universidad CEU-Cardenal Herrera, Santiago Ramón y Cajal St, 46115 Alfara del Patriarca, Valencia, Spain; 3Wildlife Ecology & Health Group (WE&H), Veterinary Faculty, Universitat Autònoma de Barcelona (UAB), Travessera dels Turons, 08193 Bellaterra, Barcelona, Spain

**Keywords:** fishborne parasites, zoonotic protozoa, *Cryptosporidium* spp., *Giardia duodenalis*, *Toxoplasma gondii*

## Abstract

In recent decades, worldwide fish consumption has increased notably worldwide. Despite the health benefits of fish consumption, it also can suppose a risk because of fishborne diseases, including parasitic infections. Global changes are leading to the emergence of parasites in new locations and to the appearance of new sources of transmission. That is the case of the zoonotic protozoa *Cryptosporidium* spp., *Giardia duodenalis*, and *Toxoplasma gondii*; all of them reach aquatic environments and have been found in shellfish. Similarly, these protozoa can be present in other aquatic animals, such as fish. The present review gives an overview on these three zoonotic protozoa in order to understand their potential presence in fish and to comprehensively revise all the evidences of fish as a new potential source of *Cryptosporidium* spp., *Giardia duodenalis*, and *Toxoplasma gondii* transmission. All of them have been found in both marine and freshwater fishes. Until now, it has not been possible to demonstrate that fish are natural hosts for these protozoa; otherwise, they would merely act as mechanical transporters. Nevertheless, even if fish only accumulate and transport these protozoa, they could be a “new” source of infection for people.

## 1. Introduction

Since 1961 fish consumption has increased at an average annual growth rate of 3.1%, overcoming average annual growth rates for all other animal proteins, except for that of poultry meat (2.7% per year). In 2017, per capita fish consumption reached a value of 20.3 kg, which represents about 17% of total animal protein. Annual per capita fish consumption varies widely between world regions, being lower in developing countries. Nevertheless, in these countries, fish represents a higher proportion of animal protein (29% in least developed countries), constituting a valuable component of their diet [1].

The rising concern about having a healthy diet to prevent diseases is one of the main reasons that has led to this increase in global fish consumption [2]. Fish composition includes different essential nutrients such as omega-3 polyunsaturated fatty acids, selenium, taurine, or vitamin D with interesting properties for human health. Moreover, fish consumption has been linked to a decrease in coronary heart disease mortality, cardiovascular disease mortality, the risk of a stroke, myocardial infarction, and heart failure. It also decreases the risk of some cancers (i.e., gastrointestinal, brain cancer, lung cancer, or multiple myeloma) and the risk of depression or cognitive impairment, among others (reviewed in [3]).

Conversely, fish, and seafood in general, can suppose a risk for human health because of the presence of physical, chemical, or biological agents. Concerning biological risk, seafood is responsible for a significant proportion of foodborne infections worldwide [4], which are caused by different viruses, bacteria, and parasites. Foodborne parasites are recognized as a neglected pathogen group despite the fact that only fish already harbor several parasite species for public health concern, including *Anisakis* spp., *Diphyllobothrium* spp., liver flukes belonging to the family Opistorchiidae or myxozoa like *Kudoa* spp., etc. [5]. The analysis of fishborne parasites is only mandatory in Europe for nematodes of the Anisakidae family, and consists of a visual inspection (Regulation (EC) No 2074/2005). This scarcity of regulation also occurs in parasites present in other food sources [6], contributing to the lack of information and control. Therefore, actual food industry policies are based on Hazard Analysis and Critical Control Points (HACCP) systems whose aim is to prevent and minimize the risk. Additionally, European laws (Regulation (EC) No 853/2004) established the obligation of freezing fishery products to be consumed raw at −20 °C for at least 24 h.

In recent years, global changes have been leading to the emergence of foodborne parasites in new locations or populations and to the appearance of new transmission routes, including new food sources [7], increasing the need for more studies and more control. This is the case of protozoan parasites, *Cryptosporidium* spp., *Giardia duodenalis*, and *Toxoplasma gondii*. Over the last two decades these protozoa have been found to enter aquatic environments and to contaminate shellfish, which appears as a new potential risk of fishborne protozoan infections [8]. This finding enhances the question of the possibility of other aquatic animals participating in *Cryptosporidium* spp., *Giardia duodenalis*, and *Toxoplasma gondii* transmission, especially fish. This review gives an overview on these three zoonotic protozoa in order to understand their potential presence in fish and to revise and discuss all the evidence of fish as a new potential source of *Cryptosporidium* spp., *Giardia duodenalis*, and *Toxoplasma gondii* transmission. It is remarkable that at the present time there is no legislation on microbiological criteria for these protozoa in any type of food. In fact, there is even a lack of standard methods for their detection on food. There is only a method for detecting *Giardia duodenalis* and *Cryptosporidium* spp. occysts on leafy green vegetables and red berry fruits (ISO 18744); this method based on immunomagnetic separation and immunofluorescence is not mandatory, but can be applied in the context of HACCP systems [6].

## 2. *Cryptosporidium* spp.

*Cryptosporidium* is a widespread protozoan parasite (Apicomplexa, Cryptosporidae) that infects a wide range of vertebrates, including humans. Cryptosporidiosis is a waterborne and foodborne parasitic disease, which is considered one of the main causes of diarrhea, especially in immunocompromised people and in children [9]. Several species have been found to infect people. The most prevalent are *Cryptpsporium hominis* (mainly anthroponotic) and *Cryptosporidium parvum* (zoonotic), but there are many more *Cryptosporidium* zoonotic species, like *Cryptosporidium meleagridis, Cryptosporium felis*, or *Cryptosporium canis*, among others [9,10]. Different genetic markers of low to moderate variability have been used to characterize *Cryptosporidium* species and, occasionally, genotypes, such as the small subunit ribosomal ribonucleic acid (SSU rRNA, also called 18S rRNA), the *Cryptosporidium* oocyst wall protein (COWP), the heat shock protein 70 (HSP70), or the internal transcriber regions (ITS). For subtyping or genetic lineage characterization, other genes, with more variability, have been used [11,12] (Table 1). It is important to highlight the 60 kDa glycoprotein (gp60) sequence analysis which has allowed the description of different subtypes for *C. hominis* (Ia-Ig), *C. parvum* (IIa-IIl), or *C. meleagridis* (IIIa-IIIf), between others. Currently, the only recognized zoonotic subtypes for *C. parvum* are the IIa and IId [11].

*Cryptosporidium* oocysts are highly resistant in the environment and are also resistant to chlorinated compounds which are commonly used in drinking and recreational water treatment [13]. Transmission can be direct via fecal–oral route or by the ingestion of contaminated water or food. *Cryptosporidium* spp. is the main agent responsible for waterborne parasitic protozoan outbreaks, including a massive outbreak in Milwaukee in 1993 with approximately 403,000 cases [14] and counting with 239 outbreaks worldwide between 2011 and 2016 [15]. Concerning the foodborne route, vegetables and fruits, especially those which are consumed raw, probably suppose the highest risk. These foodstuffs get contaminated mainly by fecally contaminated irrigation water or due to poor hygiene measures during handling and processing [16]. As a result, *Cryptosporidium* has become becomes the fifth most important foodborne pathogen [17]. *Cryptosporidium* is not only present in vegetables and fruits, but also in other foods, like dairy products, meat, or shellfish (reviewed in [18]). Many studies have demonstrated the presence of this protozoan in shellfish (i.e., [19,20,21,22,23,24,25,26,27,28]). Shellfish filter and accumulate *Cryptosporidium* oocysts from water, which proves that *Cryptosporidium* is present in different water sources, including estuarine and coastal ecosystems. Therefore, fishes could also be infected or contaminated by *Cryptosporidium* spp.

Four species have been genetically described in fish as a specific host: *Cryptosporidium molnari* in wild gilt-head sea bream (*Sparus aurata*) and European sea bass (*Dicentrarchus labrax*) [29,30], *Cryptosporidium scophtalmi* in wild turbot (*Scopthalmus maximus*) [31], *Cryptosporidium huwii* in a captive guppy (*Poecilia reticulata*) [32], and *Cryptosporium bollandi* in angelfish (*Pterophyllum scalare*) and Oscar fish (*Astronotus ocellatus*) [33] (Table 2). Furthermore, many different genotypes have been identified in fish, including piscine genotypes from two to nine, different *C. molnari*-like isolates, and different novel and un-named genotypes [34,35,36,37,38,39,40,41,42,43,44]. Interestingly, not only specific species and genotypes have been found in fish, but also other species characteristic from other hosts, including zoonotic *C. parvum*, *Cryptosporium xiaoi,* and *Cryptosporidium scrofarum* and anthroponotic *C. hominis* [36,39,44,45,46,47] (Table 3).

Therefore, the consumption of fish could be a new source of cryptosporidiosis. These findings are reviewed below, and their epidemiological significance is argued.

### Evidence of Fish as a Source of Zoonotic Cryptosporidium spp. Transmission

Various experimental studies took place in the 90s in order to ascertain if low vertebrates, namely fish, reptiles, and amphibians, could be infected by zoonotic *Cryptosporidium parvum*. In 1995, Arcay et al. [51] succeeded in infecting two fish species (*Hypostomus plecostomus* and *Hoplosternum thoracatum*), as well as different species of reptiles and amphibians, with *C. parvum* oocysts from human feces. Histological sections of experimentally infected animals revealed different phases of the *Cryptosporidium* cycle. Moreover, oocysts recovered from these animals were infectious to laboratory mice. One year later, Graczyk et al. [52] tried to bear out these results. Concerning fish, they inoculated infecting oocysts of *C. parvum* in nine bluegill sunfish (*Lepomis macrochirus*). Fish excreted oocysts in feces for up to 12 days post inoculation, but any developmental stage of the parasite was detected in gastrointestinal tissue sections. Therefore, it is probable that oocysts came from the inoculum and that fish were not truly infected. A possible explanation for the conflicting results between both studies could be the use of different *C. parvum* strains, since no molecular characterization was carried out. On the other hand, in Arcay et al. [51], fish were tested prior to the inoculation by coprological test, but this is a technique of low sensitivity, so it cannot be ruled out that these fish were previously infected with other *Cryptosporidium* species.

Freire-Santos et al. [53] conducted an experimental infection in 50 rainbow trout (*Oncorhynchus mykiss*) from a fish farm in La Coruña, Spain. Infected animals were separated in two groups according to the housing conditions: normal environmental parameters and stressful parameters. Fish in both groups showed *Cryptosporidium*-like structures in sections from stomach and pyloric region, the number of these structures being higher in animals under stressful conditions. Structures were tested positive against anti-*C. parvum* antibodies by immunofluorescence antibody test (IFAT). Even though these results seem to be in accordance with the Arcay et al. study [51], IFAT can cross react with other *Cryptosporidium* species and, in this case, rainbow trout were also tested only by coprological test.

The question of whether fish could be infected by zoonotic *C. parvum* or they simply retain and disseminate the oocysts remained without a clear answer. Years later, field studies began to be carried out. Reid et al. [36] analyzed gastrointestinal tissue scrapings from 227 cultured freshwater fingerlings, 255 wild marine fish, and 227 wild freshwater fish from Australia. Only 0.8% of fish (6/709) were positive for *Cryptosporidium* spp., all of them marine fish (2.4%; 6/255). One positive isolate from a school whiting (*Sillago vittata*) was genetically identical to *C. parvum* at both 18S rRNA and actin loci, subtyping at the gp60 locus as the subtype IIaA18G3R1. Subfamily IIa is the most common in calves and humans in North America, Europe, and Australia [54]. This was the first description of *C. parvum* in a fish not experimentally infected. Nonetheless, no histopathological examination was performed, so it was not possible to know if this was a true infection or a mechanical transport. Furthermore, another isolate was identified as *C. xiaoi* and two others as pig genotype II (now *C. scrofarum*). Currently, both species are also considered as zoonotic (reviewed in [32]).

Gibson-Kueh et al. [55] carried out a nested polymerase chain reaction (PCR) of 18S rRNA gene on 20 previously fixed tissues of cultured barramundi (*Lates calcarifer*) from Vietnam. Partial sequences were obtained from two formalin fixed tissue samples, which were closely related with *C. parvum* type B.

Similarly to [36], Koinari et al. [39] worked with three fish groups: 133 cultured fish, 205 wild freshwater fish, and 276 wild marine fish from Papua New Guinea. The genetic analysis of intestinal and stomach scrapings revealed seven positives between the three groups. Five isolates were identified as *C. parvum* at the 18S rRNA locus. At the gp60 locus, four of them subtyped in the IIa subfamily and the fifth was identified as *C. hominis* (mixed infection). Histopathological examination failed because of autolysis, so it was not possible to discern between a natural infection and a mechanical transport.

Certad et al. [45] analyzed 41 adult fish from Lake Geneva in France. In total, 37% (15/41) and the prevalence of *Cryptosporidium* was detected. Sequences of positive animals correspond to *C. parvum* (*n* = 13), *C. molnari* (*n* = 1), and *C. parvum–C. molnari* mixed infection (*n* = 1). Three different subtypes were identified at gp60 locus, all of them pertaining to the IIa subfamily. Histological analysis suggests a natural infection for the first time; intracellular *Cryptosporidium*-like bodies were found in 10 of 13 *C. parvum* isolates.

Couso-Pérez et al. [43] performed a study on 360 rainbow trout from a farm in Santiago de Compostela, Spain. In this case, aliquots of sediments from pyloric caeca and intestinal content were used in the analysis. A total of 33 samples (9.2%) were positives by IFAT. Eleven partial sequences of the SSU-rDNA were obtained and *C. parvum* was identified in seven of them. It was not possible to amplify gp60 and actin loci. The detection of a cluster of 10 oocysts in the pyloric caeca of one animal could indicate a true infection.

Another study by the same authors [47] analyzed the presence of *Cryptosporidium* in 613 brown trout (*Salmo trutta*) from Galician (NW Spain) rivers. Same samples (pyloric caeca and intestine) were used, resulting in 103 positive individuals by IFAT. Among these positives, *C. parvum* was molecularly identified in 47 fish (subtypes IIaA15G2R1 and IIaA18G3R1). Oocyst clusters were observed in five cases, suggesting a possible true infection.

Certad et al. [44] carried out two studies in France. A regional survey that was composed of 1508 fish from 31 different species showed a prevalence of 2.3% (35/1508). Eight of them were identified as *C. parvum* with four subtypes of the subfamily IIA represented. It was possible to identify compatible structures within the epithelial cells and in an apical position in just one case. The second study, at Boulogne-Sur-Mer, analyzed 345 fish. A prevalence of 3.2% (11/345) was detected, with two isolates identified as *C. parvum* (subtyping could not be attained). This is the first epidemiological and molecular data on fish cryptosporidiosis in European waters.

The latest evidence comes from Shahbazi et al. [46] who investigated the presence of zoonotic *C. parvum* in freshwater ornamental fish from different pet shops in Iranian Iran cities. Sixteen out of 100 animals were positive by histopathology; only two samples, corresponding to goldfish (*Carassius auratus*) were positive for *C. parvum* by nested PCR at the 18S rRNA gene [56]. One goldfish presented the intestine with emaciation, while the other was clinically healthy. Therefore, it was not possible to assess if it is a true or a mechanical infection.

Molecular methods used in the different mentioned studies are summarized in Table 4. Gastrointestinal tissue scrapings are the most common sample used for the analysis of *Cryptosporidium* in fish. Nested PCR, and subsequent amplification products sequencing, has been the major technique used in all cases, 18S rRNA and gp60 being the most common target genes.

Although some studies provide histopathological evidence of natural infection in fish hosts, this cannot be confirmed at present. However, in any case, fishes would be acting as mechanical vectors of zoonotic *Cryptosporidium* spp., contributing to the dissemination of oocysts in water and being themselves a potential source of human cryptosporidiosis. Currently, there is no evidence of fish to human transmission, but one study reports that urban anglers are at high risk of contracting *C. parvum* infection while fishing and consuming fish [57].

## 3. *Giardia duodenalis*

Giardiasis caused by *Giardia duodenalis* (Diplomonadida, Giardiidae) is a very prevalent disease worldwide, responsible for approximately 280 million cases of diarrhea every day [58]. Currently, eight species of *Giardia* exist, but only *G. duodenalis* can infect humans and, therefore, it is the only one considered as zoonotic [59]. *G. duodenalis* is a species complex with eight different assemblages (A-H) recognized [60]. Although assemblages A and B have been traditionally considered the only ones that can infect humans, assemblages C, D, E, and F have also been identified in people [61] (Table 3). Finally, assemblage A is subdivided into three different subtypes (AI-AIII), subtype AII being the most commonly detected in humans [62]. Recently, some authors have proposed to reconsider the different *G. duodenalis* assemblages as a different species based on traditional taxonomic descriptions and biological differences, mainly host specificity [12,63] (Table 5). Similarly to *Cryptosporidium*, different genes have been used to characterize *Giardia* species and genotypes, including the SSU rRNA, the ITS, the *Giardia* specific *β-giardin*, the triosephosphate isomerase (TPI), the glutamate dehydrogenase (GDH), or the elongation factor 1 (ef1) genes [12] (Table 6).

Cysts of *Giardia* are environmentally resistant. Therefore, transmission occurs directly via fecal–oral or through consumption of contaminated water or food [62]. There are many documented outbreaks of waterborne giardiasis [74], however foodborne published outbreaks of giardiasis are few (probably under-reported). Even so, *G. duodenalis* is the 11th foodborne parasite according to the World Health Organization [17] (*Cryptosporidium* occupies the fifth position). The foodstuffs most commonly implicated in these foodborne outbreaks are fresh products [74], particularly those consumed raw or undercooked. Similar to *Cryptosporidium*, *Giardia* cysts arrived on fruits and vegetables by fecal-contaminated irrigation water or by handlers with poor hygienic habits [75].

*G. duodenalis* is present in different water sources, which become contaminated by human or farm animal discharges, or even by wildlife feces [8]. Like *Cryptosporidium*, *G. duodenalis* arrives at marine environments and bioaccumulates in shellfish [20,21,26,27,76,77,78,79,80,81,82,83,84,85]. It would be possible that fish become infected by zoonotic *G. duodenalis* (due to the broad spectrum of hosts of this parasite) or, in other cases, they could act as a mechanical transmitter (like shellfish are). Therefore, potentially, the consumption of fish could be a new source of giardiasis in humans. In the next section, we review and discuss the potential role of fish as a source of *G. duodenalis* transmission.

### Evidence of Fish as a Source of Zoonotic Giardia duodenalis Transmission

Lassek-Neselquist et al. [86] performed the first study on *G. duodenalis* molecular epidemiology in marine vertebrates. In the context of this study, *G. duodenalis* was reported for the first time in feces and gut content of a single analyzed fish, specifically in a thresher shark (*Alopias vulpinus*). The authors identified a human-infecting *G. duodenalis* haplotype of assemblage B in this individual; this haplotype corresponded to a GDH sequence.

Nonetheless, the first study aimed at the detection of different species or assemblages of *Giardia* in fish was conducted two years later in Australia by Yang et al. [87]. In this study three groups of fish were analyzed, cultured fingerlings (*n* = 227), wild freshwater fish (*n* = 227), and wild marine fish from coastal and estuarine areas (*n* = 255). Analysis was performed on intestinal and stomach tissue scrapings and consisted of four different loci analysis by PCR and a histological examination. In summary, 27 of the 709 fish (3.8%) were positive, 8.4% of the cultured fingerlings (19/227; *Lates calcarifer*, *Acanthopagrus butcheri*, *Argyrosomus japonicus,* and *Pagrus auratus*), 2.7% of the wild marine fish (7/255; *Mugil cepahalus*), and 0.4% of the wild freshwater fish (1/227; *Galaxias occidentalis*). *G. duodenalis* assemblages A, B, and E, in addition to *G. microti* were identified. It is important to remark that *Giardia* cysts and trophozoites were detected in large numbers in 10 positive samples. This finding, together with the fact that analyses were conducted on tissue scrapings (instead of feces) seems to demonstrate that fish are truly infected, and they are not mere mechanical vectors.

Given the detection of zoonotic genotypes in fish [87], Ghoneim et al. [88] carried out another study in fish in Egypt. They analyzed a total of 92 animals, fish from farms and wild fish from the Nile River. In this case, they obtained fecal samples instead of intestinal and stomach tissue scrapings. Firstly, they used an enzyme-linked immunosorbent assay (ELISA) to screen for positive samples and secondly, they performed a specific duplex PCR on positive samples for identification of zoonotic assemblages A and B [89]. In this study, the only gene amplified was the TPI gene. Three fish were tested positive (3.3%), one farmed fish (*Tilapia nilotica*) and two wild sea mullets (*Mugil cephalus*). In all three cases, assemblage A was identified.

Table 7 summarizes molecular approaches used in these three studies. Two of them have used feces (or feces and gut content) as choice sample for *Giardia duodenalis* detection. Similarly, to *Cryptosporidium*, nested PCR, and subsequent sequencing, has been the most common technique. The most frequent targeted genes have been *β-giardin*, GDH, and TPI ones.

Since then, no further studies have been conducted on the occurrence of zoonotic *G. duodenalis* assemblages in fish. The only posterior reference was that of Nematollahi et al. [90]. These authors carried out a study on histopathological detection of endoparasites in freshwater ornamental fish in Iran. In this study, *Giardia* was observed in the intestine of one Oscar fish (*Astronotus ocellatus*). However, no molecular identification was performed.

Based on the study of Yang et al. [87], Tysnes et al. [91] decided to try an experimental model on the use of zebrafish (*Danio rerio*) as a *G. duodenalis* infection model. Except for assemblage A, other *G. duodenalis* assemblages seem to be refractory at being cultivated in vitro or to being established in laboratory animals [92]. The likely *Giardia* infection in wild and cultivated fish detected by Yang et al. [87] could be the solution, providing a novel animal model. Zebrafish were inoculated with viable cysts of *G. duodenalis* assemblages A and D and the assessment of the infection was made by different methods (direct microscopy, IFAT, direct cultivation, and histology). *Giardia* was not detected by direct microscopy or histology, and direct cultivation was also not possible. Despite this, IFAT demonstrated the retention of cysts in the intestine, even some days after the inoculation. However, the number of cysts was decreasing. Therefore, the results were inconclusive, and more research is needed to establish if there could be long term infection by *G. duodenalis*.

## 4. *Toxoplasma gondii*

*Toxoplasma gondii* (Apicomplexa, Sarcocystidae) is a well-known zoonotic parasite all over the world. It affects virtually all warm-blooded animals and both terrestrial and aquatic species. It is a widespread parasite of high concern for in public health which is the cause of severe illness and even death in the case of acquired toxoplasmosis in immunocompromised patients. It also causes spontaneous abortion and congenital birth defects when the infection is acquired during pregnancy [93]. It has been calculated that approximately 30% of the world’s population is infected [94]. *Toxoplasma gondii* is a single species which mainly consists of three clonal lineages (type I, II, and III strains). Type I strains are known to cause acute mortality in murine models, while types II and III, lead to chronic infections. In Europe and North America, type II and, to a lesser extent, type III strains, are the most prevalent. Furthermore, in North America, an atypical sylvatic strain circulates in wildlife (haplogroup 12). Conversely, in South America, the genetic diversity is much higher, with several atypical strains having been identified [95].

Intermediate hosts, including humans, can be infected by three routes: vertical transmission from mother to fetus (not discussed here), ingestion of tissue cysts from infected animals, and ingestion of oocysts through contaminated water, soil, or food [96]. It must be remarked upon that the only definitive host able to eliminate infecting oocysts are felids, with special attention given to domestic and feral cats (*Felis silvestris catus*).

Traditionally, most human infections have been considered to be acquired by the ingestion of raw or undercooked meat containing cysts [97], however, sanitary improvements in animal production and the food-processing industry have reduced this associated-risk [98]. Unlike *G. duodenalis* and *Cryptosporidium* spp., only two outbreaks have been associated with fresh products contaminated with oocysts [99,100].

Until recently, environmental transmission had been the less studied route, despite the high resistance of *T. gondii* oocyst [101]. *Toxoplasma* has come to be considered as a significant waterborne parasite, only in the last few decades when waterborne outbreaks by the consumption of water containing oocysts have been detected [102,103,104,105,106]. Several sea mammals have been found to be infected by *T. gondii* [107], setting up evidence that transmission between terrestrial and marine environments takes place and highlighting the potential significant role of waterborne transmission.

Aquatic environments, including estuarine and marine ones, get polluted by sewage and freshwater runoffs carrying *T. gondii* oocysts [108]. In coastal environments, oocysts can be accumulated by bivalves. This bioaccumulation has been demonstrated experimentally [109,110,111] and has been detected in wild and cultivated bivalves [24,25,26,85,112,113,114,115,116,117,118,119,120,121]. Nonetheless, this cannot explain how offshore cetaceans that do not feed on bivalves come into contact with these parasitic oocysts and acquire the infection. Alternative pathways have been proposed, for instance, the marine macroaggregates, which are associations of micropolymers with bacterial biofilm, algae, and other microorganisms [84]. Another possibility is that fish, not only shellfish could act as mechanical carriers of *T. gondii* oocysts, disseminating these during migration [122]. This possibility implies a new potential source of *T. gondii* transmission to humans, by the consumption of raw or undercooked contaminated fish.

### Evidence of Fish as a Source of Zoonotic Toxoplasma gondii Transmission

In 2005, Omata et al. [123] carried out the first attempt to test if *T. gondii* were be able to multiply and persist in cells of a cold-blooded host, the goldfish (*Carassius auratus*). Tachyzoites of the Beverley (type II) and RH (type I) strains penetrated and multiplied in oviduct epithelial cells of goldfish in vitro at 37 °C. However, at 33 °C no tachyzoite penetrated into cells. Additionally, an in vivo inoculation test was conducted at 37 °C by injecting tachyzoites intramuscularly in goldfish. Fish were euthanized on days three and seven post inoculation and *T. gondii* detection was performed by DNA detection (*T. gondii* B1 gene, [124]) and mice bioassay. In both cases, the parasite was detected on day three, but not on day seven. Based on these authors’ results, *T. gondii* is not transmissible to freshwater fish.

Taghadosi et al. [125] used a different approach to check if fish, in this case, farmed salmonids in Iran, could be primarily infected by *T. gondii*. They obtained 50 sera samples and tested them for *T. gondii* specific IgM and IgG by ELISA. All samples were negative for IgG, but five of 50 samples were positive for IgM, indicating a possible recent infection in these fishes.

Similarly to the case of *G. duodenalis*, Sanders et al. [126] tried a new in vivo model of *T. gondii* infection using the zebrafish. Fish kept at 37 °C were intraperitoneally injected with tissue cysts bradyzoites from experimentally infected mice; two strains, corresponding to types II and III were used. *T. gondii* tachyzoites were observed up to seven days post injection in several fish tissues, which had clinical signs before they died or were euthanized. This experiment was repeated using cysts from mice brains and cultured tachyzoites and the same results were obtained. These authors were able to establish a successful experimental *T. gondii* acute infection in a cold-blooded host, the zebrafish, for the first time.

The opposite results obtained by Omata et al. [123] in goldfish and Sanders et al. [126] in zebrafish were able to rely on a different experimental approach: different host species, different inoculation site (intramuscular and intraperitoneal, respectively), and a different method of analysis (cytology, PCR and mouse assay vs. histopathology). Nonetheless, in both cases, high temperature (37 °C) seems to be an important limiting factor. In agreement with this finding, Yang et al. [127] showed that *T. gondii* invasion and replication in fish cells (in vitro study) is temperature-dependent: the minimum temperature for tachyzoites penetration is 27 °C, while a minimum of 30 °C is necessary for their replication. Furthermore, Yoshida et al. [128] have been successful in using the zebrafish as a novel model in vivo for studying *T. gondii* replication and interaction with macrophages. On the other hand, results from Sanders et al. [126] are in agreement with the findings of Taghadosi et al. [125], since both provide evidence of *T. gondii* primary infection in fish.

Previous studies seem to show contradictory or unclear results about the possibility of *T. gondii* infecting cold-blooded hosts, such as fish. For the moment, there is not enough evidence and further research is needed, especially concerning studies on wild fish populations instead of experimental infection studies.

In spite of this, even if fish could not be naturally infected by *T. gondii*, they could still be a source of infection by accumulating and spreading oocysts, just as shellfish do [109,110,111]. To test the hypothesis of fish acting as biotic vectors for *T. gondii*, Massie et al. [122] exposed two migratory filter-feeding marine fish species, northern anchovies (*Engraulies mordax*) and Pacific sardines (*Sardinops sagax*), to *T. gondii* oocysts. Fish were euthanized at 2 and 6–8 h after exposure and their alimentary tracts were recovered and analyzed. Two hours post-exposure *T. gondii* was detected by PCR in 75% of northern anchovies and 60% of Pacific sardines, while 6–8 h after exposure 50% of the northern anchovies were positive. Concerning mice bioassay, 30% of mice fed samples from fish exposed to 100,000 oocysts/L manifest compatible clinical signs and were tested positive for *T. gondii* B1 gene. None of fish exposed to lower oocyst concentration became infected. Thus, both migratory filter-feeding species are able to filter *T. gondii* oocysts and to retain these oocysts infective, for at least 8 h, indicating that marine fish can act as mechanical vectors. Nonetheless, these results were obtained under experimental conditions and for a short-period duration.

In 2014, Zhang et al. [116] conducted a field study to detect *T. gondii* in the digestive tissues of cultured fish and shellfish in China. Four fish species were included: 309 *Carassius auratus*, 98 *Momopterus albus*, 456 *Hypophthalmichthys molitrix,* and 309 *Cyprinus carpio.* Only one *H. molitrix* (0.8%) tested positive by PCR for *T. gondii* ITS-1 region, which is markedly conserved [129]. This was the first study to find *T. gondii* in naturally exposed fish.

Aakool and Abidali [130] carried out a study to detect *T. gondii* in different tissues (muscle, liver, intestine, and gills) of freshwater *Cyprinus carpio* from the Tigris River (Iraq). Sixteen out of 24 samples were positive by real time PCR (rt-PCR; *T. gondii* B1 gene [131]) for *T. gondii* in the intestine, as well as two gill samples.

More recently, Marino et al. [98] carried out a large field study in Sicily, comprising 1293 edible marine fish species from extractive fisheries. These authors not only analyzed gastrointestinal tissues, but also gills and skin-muscle samples. For example, gills could retain *T. gondii* filtered from seawater and skin-muscle could be contaminated by contact with sea-beds, for example. Samples were pooled by species and sample type in 147 groups. Of these, 32 samples (21.8%) tested positive on rt-PCR [132]; positive samples were detected in 12 of 17 species and in all three-types of samples. Among the positive species, there was heterogeneity concerning the type of habitat and dietary habits. These findings provide more evidence of the possible role of marine fish in the transmission of *T. gondii* and, therefore, the emergence of a possible new seafood disease.

A summary on molecular methods used in *Toxoplasma gondii* detection in fish is shown in Table 8. Unlike *Cryptosporidium* and *Giardia duodenalis*, a broad range of samples have been analyzed in these studies, highlighting muscle and intestine. In this case, rtPCR and subsequent sequencing, has been the most applied technique. Genes used to molecularly identify *Toxoplasma gondii* are mainly two: *T. gondii* B1 gene and a 529-bp DNA repeat element from the parasite genome (GenBank accession no. AF146527).

## 5. Conclusions

The studies discussed above do not allow us to determine whether fish can be naturally infected by protozoa *Crypstosporidium* spp., *Giardia duodenalis,* and/or *Toxoplasma gondii* or they simply act as mechanical transporters. Nevertheless, even if fish are only mechanical carriers, they could be a “new” source of infection for people. Field studies demonstrate that all these three zoonotic protozoa are present both in freshwater and marine fishes, including species of commercial interest. Although fish seem to accumulate the parasitic forms mainly in the gastrointestinal tract, which is not consumed, there is still a risk of infection because of cross-contamination of the fish meat during evisceration and processing. This potential new risk could increase the list of fishborne parasitic infections, necessitating more studies to achieve a complete understanding of the possible role of fish in human cryptosporidiosis, giardiasis, and toxoplasmosis.

## Figures and Tables

**Table 1 foods-09-01913-t001:** Target genes used in *Cryptosporidium* spp. characterization. Adapted from [11,12].

Target Gene	Gene Copies ^1^	Level of Characterization	Species Range	Amplification Success ^2^
SSU rRNA (18S rRNA)	Multiple	Species, genotypes	All	Great
ITS	Multiple	Species, genotypes	*C. parvum*, *C. hominis*, and close-related	Good
HSP70	Single	Species, genotypes	*C. parvum*, *C. hominis,* and close-related	Good
COWP	Single	Species, genotypes	*C. parvum*, *C. hominis,* and close-related	Variable
gp60	Single	Species, subgenotypes	*C. parvum*, *C. hominis,* and close-related	Variable

^1^ The target gene is present in a single copy (Single) or in multiple copies (Multiple) in the haploid genome of the parasite; ^2^ In general, multiple copies genes have better amplification success in polymerase chain reaction (PCR) techniques.

**Table 2 foods-09-01913-t002:** Fish-specific *Cryptosporidium* species and their hosts.

Species	Host Fish Species	References
*C. molnari*	*Sparus aurata* *Dicentrarchus labrax*	[29,48]
	*Maccullochella peelii*	[49]
	*Esox lucius*	[45]
*C. scophthalmi*	*Scophthalmus maximus*	[31]
	*Psetta maxima*	[50]
*C. huwi*	*Poecilia reticulata*	[32,40]
	*Puntigrus tetrazona*	[40]
	*Paracheirodon innesi*	[37,40]
*C. bollandi*	*Pterophyllum scalare* *Astronotus ocellatus*	[33]

**Table 3 foods-09-01913-t003:** Other *Cryptosporidium* species detected in fish hosts.

Species	Host Fish Species	References
*C. parvum*	*Sillago vittata*	[36]
	*Decapterus macarellus* *Puntius gonionotus* *Oreochromis niloticus*	[39]
	*Salvelinus alpinus* *Esox lucius* *Coregonus lavaretus* *Perca fluviatilis* *Rutilus rutilus*	[45]
	*Oncorhynchus mykiss*	[43]
	*Salmo trutta*	[47]
	*Molva dypterygia* *Gadus morhua* *Scomber scombrus* *Scomber japonicus* *Sardina pilchardus* *Engraulis encrasicolus* *Clupea hareguns*	[44]
	*Carassius auratus*	[46]
*C. hominis*	*Decapterus macarellus*	[39]
*C. scrofarum*	*Sillago vittata*	[36]
*C. xiaoi*	*Sillago vittata*	[36]

**Table 4 foods-09-01913-t004:** Comparative table on the molecular methods (technique and target genes) used for identifying zoonotic *Cryptosporidium* species in fish samples.

Study	Sample	Molecular Technique	Target Genes
Reid et al., 2010 [36]	Gastrointestinal tissue scrapings	Nested PCR	18S rRNA Actin locus gp60
Gibson-Kueh et al., 2011 [55]	Fixed intestinal tissue	Nested PCR	18S rRNA Actin locus
Koinari et al., 2013 [39]	Gastrointestinal tissue scrapings	Nested PCR-RFLP Nested PCR Seminested PCR	18S rRNA gp60 Actin locus
Certad et al., 2015 [45]	Gastrointestinal tissue scrapings	Nested PCR	18S rRNA gp60
Couso-Pérez et al., 2018 [43]	Pyloric caeca Intestinal content	Nested PCR	18S rRNA gp60 HSP70 Actin locus
Couso-Pérez et al., 2019 [47]	Pyloric caeca Intestine	Nested PCR	18S rRNA gp60 HSP70 Actin locus
Certad et al., 2019 [44]	Gastrointestinal tissue scrapings	Nested PCR	18S rRNA gp60
Shahbazi et al., 2020 [46]	Stomach and intestine tissues	Nested PCR	18S rRNA

**Table 5 foods-09-01913-t005:** Recognized *Giardia* species, their host range, and zoonotic potential.

*Giardia* Species	*Giardia* Species According to [12,63]	Hosts	Zoonotic (Z) or Reported in Humans (RH)
*G. duodenalis* assemblage A	*G. duodenalis*	Humans, other primates, and a wide range of domestic and wild mammals	Z
*G. duodenalis* assemblage B	*Giardia enterica*	Humans, other primates, dogs, and some species of mammalian wildlife	Z
*G. duodenalis* assemblage C	*Giardia canis*	Canids, including dogs	RH [64,65]
*G. duodenalis* assemblage D	*Giardia canis*	Canids, including dogs	RH [66]
*G. duodenalis* assemblage E	*Giardia bovis*	Cattle and other hoofed animals	RH [67,68,69,70,71,72]
*G. duodenalis* assemblage F	*Giardia cati*	Cats	RH [73]
*G. duodenalis* assemblage G	*Giardia simondi*	Rats	–
*G. duodenalis* assemblage H		Marine mammals	–
*Giardia agilis*	*G. agilis*	Amphibians	–
*Giardia muris*	*G. muris*	Rodents	–
*Giardia psittaci*	*G. psittaci*	Birds	–
*Giardia ardeae*	*G. ardeae*	Birds	–
*Giardia microti*	*G. microti*	Microtine voles and muskrats	–
*Giardia cricetidarum*	*G. cricetidarum*	Hamsters	–
*Giardia paramelis*	*G. paramelis*	Southern brown bandicoots	–

**Table 6 foods-09-01913-t006:** Target genes used in *Giardia* spp. characterization. Adapted from [12].

Target Gene	Gene Copies ^1^	Level of Characterization	Amplification Success ^2^
SSU rRNA (18S rRNA)	Multiple	Species	Great
ITS	Multiple	Species, subgenotypes	Good
*β-giardin*	Single	Species, subgenotypes	Variable
TPI	Single	Species, subgenotypes	Variable
GDH	Single	Species, subgenotypes	Variable
ef1	Single	Species, subgenotypes	Variable

^1^ The target gene is present in a single copy (Single) or in multiple copies (Multiple) in the haploid genome of the parasite; ^2^ in general, multiple copies genes have better amplification success in polymerase chain reaction (PCR) techniques.

**Table 7 foods-09-01913-t007:** Comparative table on the molecular methods (technique and target genes) used for identifying zoonotic *Giardia duodenalis* assemblages in fish samples.

Study	Sample	Molecular Technique	Target Genes
Lasek-Nesselquist et al., 2008 [86]	Feces Gut content	Nested PCR	*β-giardin* GDH TPI mlh1
Yang et al., 2010 [87]	Gastrointestinal tissue scrapings	Nested PCR	18S rRNA *β-giardin* GDH TPI
Ghoneim et al., 2012 [88]	Feces	Duplex PCR	TPI

**Table 8 foods-09-01913-t008:** Comparative table on the molecular methods (technique and target genes) used for identifying *Toxoplasma gondii* in fish samples.

Study	Sample	Molecular Technique	Target Genes
Omata et al., 2005 [123]	Muscle Brain Liver Kidney	PCR	*T. gondii* B1 gene
Massie et al., 2010 [122]	Oocysts recovered from alimentary canals	rt-PCR PCR	*T. gondii* B1 gene 529-bp DNA repeat element
Zhang et al., 2014 [116]	Digestive tract	PCR	ITS-1
Aakool and Abidali, 2015 [130]	Muscle Liver Intestine Gills	rt-PCR	*T. gondii* B1 gene
Marino et al., 2019 [98]	Intestine Gills Skin-skeletal muscle	rt-PCR qPCR	529-bp DNA repeat element

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
