# Peer review of "Potential Risk of Three Zoonotic Protozoa (Cryptosporidium spp., Giardia duodenalis, and Toxoplasma gondii) Transmission from Fish Consumption"

_foods, 2020, doi:10.3390/foods9121913_

Round 1

Reviewer 1 Report

Manuscript ID: foods-1034879 The review manuscript is an overview on these three zoonotic protozoa in order to understand their potential presence in fish and to comprehensively revise all the evidences of fish as a new potential source of Cryptosporidium spp, Giardia duodenalis and Toxoplasma gondii transmission which normally could not demonstrates these fish as hosts for protozoa or mechanical transporters to human infection. In overall the manuscript has provided such a great contain and well organization of the knowledge.

However, the Figures and some expressions are hard to understand for the readers. Therefore, I recommend the authors thoroughly checked and changed the style of expression the data, described texts as much as possible to improve the quality of the papers.

Major’s comments 1- The data shows in Figs. 1&2 are not well organized which makes it hard to understand for the readers. What is the exactly meaning of the term “Multiple- gene copy number and Single- gene copy number”; amplification success? The authors are suggested to re-arrange the data become table style or figure style with more neat and clear expression and meaning of this data.

2- The authors suggested adding some tables which shows the methods used for identifying the fish by molecular technique for easing eye-catching and comparison. Since there is a lot of methods were used as shown in the manuscript.

Minor’s comments 1- The author’s addresses should be written in English.

2- In the manuscript, there is a lot words have the same expression but having different styles of writing (fish borne (line 28) vs fishborne (lines 16, 53, 65, 418; 37 C vs 37 C; ). These mistakes should be avoided by using the identical expression.

3- Some abbreviations are not show full expression for the first time (PCR: there is no “Polymerase chain reaction (PCR)”; rt-PCR). The authors should thoroughly check the manuscript and corrected these mistakes for easily understanding from readers viewpoints.

4- The references styles are not similar such as DOI number’s information, page number, etc. Therefore the authors are suggested to correct and fix all references as journal guideline.

5- The manuscript should be thoroughly checked English since there are a lot of sentences which hard to understand and follow the meaning. Example, (line 139) “One year later, Graczyk et al. [52] try to bear out these results” => “In 1996, Graczyk et al. …”; (line 172) “Similar to [38]” => “Similar to Reid et al.”. Also there were a lot of grammatical errors which must be avoided.

Author Response

Valencia, December 15th, 2020

To,

Editor of Foods

Dear Editor,

I am sending the revised version of the manuscript entitled “Potential risk of zoonotic protozoa transmission from fish consumption: a review on Cryptosporidium spp., Giardia duodenalis and Toxoplasma gondii”, intended for publication in Foods as a review paper.

Please find attached to this cover letter a revised version of the manuscript, thoroughly prepared to accommodate the comments and recommendations made by the Reviewers.    
For your convenience, the detailed list of all the changes, together with the answers to the comments raised by the Reviewers, is provided bellow.

Response to Referee 1 comments:

We sincerely thank the reviewer for the constructive comments and suggestions, which helped us to substantially improve our manuscript. Please find the point-to-point responses (blue) to the comments (black) as listed below.

Major’s comments:

  1. The data shows in Figs. 1&2 are not well organized which makes it hard to understand for the readers. What is the exactly meaning of the term “Multiple-gene copy number and Single-gene copy number”; amplification success? The authors are suggested to re-arrange the data become table style or figure style with more neat and clear expression and meaning of this data.

Following the suggestion, data on figures 1 and 2 have been re-arranged in table style (now, Tables 1 and 6). In this way, as you indicated, we think that the information is easier to rapidly understand. Moreover, two foot notes have been included in these tables for the better understanding of gene copies and amplification success concepts.

  1. The authors suggested adding some tables which shows the methods used for identifying the fish by molecular technique for easing eye-catching and comparison. Since there is a lot of methods were used as shown in the manuscript.

Three new tables have been included in subsections 2.1, 3.1 and 4.1 (Tables 4, 7 and 8, for Cryptosporidium spp, Giardia duodenalis and Toxoplasma gondii, respectively). Each one of these three tables summarizes the molecular methods used in the different studies concerning these protozoa in fish (the studies mentioned and discussed in the main text). Type of sample, molecular technique and target genes are shown for each study, facilitating a rapid overview and comparison for the readers. Additionally, three new paragraphs have been included in each of these three sections in order to introduce the tables and extract the main information: lines 214-217, 290-293 and 428-433.

*As a result of these changes, table numeration has been re-arranged throughout the text and in table titles.

Minor’s comments:

  1. The author’s addresses should be written in English.

Author’s addresses have been re-written in English, except for the group name in address 1 (Servicio de Análisis, Investigación y Gestión de Animales Silvestres (SAIGAS). We think the original name can facilitate the search of our group by other researchers in scientific data basis.

  1. In the manuscript, there is a lot words have the same expression but having different styles of writing (fish borne (line 28) vs fishborne (lines 16, 53, 65, 418; 37 °C vs 37° C; ). These mistakes should be avoided by using the identical expression.

Writing styles for these expressions have been homogenised, as well as other two mistakes that we have detected when checking the document:

  1. Fish borne (now line 32) has been corrected as fishborne, like in other cases.
  2. All temperature expressions have been expressed in the following format: X ºC. Two changes have been made in line 363.
  3. In lines 272-273 expressions like n=227 have been rewritten as N = 227, in order to follow the format in lines 185-186.
  4. Finally, in line 211, Nested-PCR has been corrected (nested PCR). In this way, it coincides with the same expression in other parts of the text and tables.
  1. Some abbreviations are not show full expression for the first time (PCR: there is no “Polymerase chain reaction (PCR)”; rt-PCR). The authors should thoroughly check the manuscript and corrected these mistakes for easily understanding from readers view points.

Abbreviations used in the manuscript have been checked. The following changes have been made to correct the mistakes:

  1. Line 61: full expression for HACCP (Hazard Analysis and Critical Control Points) has been added.
  2. Lines 89-90: full expression for SSU rRNA (small subunit ribosomal ribonucleic acid) has been added.
  3. Line 173: full expression for PCR (polymerase chain reaction) has been added.
  4. Line 285: full expression for ELISA (enzyme-linked immunosorbent assay) has been added here, as it is the first mention, and eliminated from the second mention in line 371.
  5. Line 417: full expression for rt-PCR (real time PCR) has been added here, as it is the first mention, and eliminated from the second mention in line 423.
  1. The references styles are not similar such as DOI number’s information, page number, etc. Therefore the authors are suggested to correct and fix all references as journal guideline.

We have revised all references and all the errors in the style have been corrected according to the journal guideline. Specially, as you pointed out, we have corrected page numbers and doi format, following journal instructions.

  1. The manuscript should be thoroughly checked English since there are a lot of sentences which hard to understand and follow the meaning. Example, (line 139) “One year later, Graczyk et al. [52] try to bear out these results” => “In 1996, Graczyk et al. …”; (line 172) “Similar to [38]” => “Similar to Reid et al.”. Also, there were a lot of grammatical errors which must be avoided.

Finally, and following your recommendation, the manuscript has been checked in English by a native professional, taking into account your comments. We can provide the certificate of their correction if it is necessary.

Reviewer 2 Report

This manuscript presents a comprehensive biliographic review of zoonotic protozoa present in fish. It seems to me an interesting contribution, especially because of the exhaustive review they provide on evidence that shows how fish, both freshwater and marine, compiling more than 130 citations corresponding to works from the last decade.

However there are some minor issues:

Table 1 lists the protozoa as well as the fish species that are potential hosts with the corresponding references. It would be interesting if these references were ungrouped according to the species of fish instead of grouped according to the species of protozoa. Similar consideration is applicable to table 2

Author Response

Valencia, December 15th, 2020

To,

Editor of Foods

 Dear Editor,

I am sending the revised version of the manuscript entitled “Potential risk of zoonotic protozoa transmission from fish consumption: a review on Cryptosporidium spp., Giardia duodenalis and Toxoplasma gondii”, intended for publication in Foods as a review paper.

Please find attached to this cover letter a revised version of the manuscript, thoroughly prepared to accommodate the comments and recommendations made by the Reviewers.    
For your convenience, the detailed list of all the changes, together with the answers to the comments raised by the Reviewers, is provided bellow.

Response to Referee 2 comments:

We sincerely thank the reviewer for the constructive comments and suggestions, which helped us to substantially improve our manuscript. Please find the point-to-point responses (blue) to the comments (black) as listed below.

  • Table 1 lists the protozoa as well as the fish species that are potential hosts with the corresponding references. It would be interesting if these references were ungrouped according to the species of fish instead of grouped according to the species of protozoa. Similar consideration is applicable to table 2.

Tables 1 and 2 (now tables 2 and 3 because of changes suggested from other referee) have been modified in order to group the references by Cryptosporidium sp. and, also, by fish species. In this way, we combine the original information with that you have suggested, showing the information as detailed as possible. We think it is interesting to also maintain Cryptosporidium species grouping because, in most cases, each fish only harbour one Cryptosporidium species.  

Reviewer 3 Report

This manuscript basically reviews three zoonotic protozoa, their presence in the fish, as well as the potential risk from the fish consumption. The structure of this review is pretty straightforward. In each section, it started from the brief introduction of the specific protozoa, and followed by the evidence of its presence in the fish. In the end, it was concluded there is still not clear that the presence of the protozoa in the fish is because of naturally infection or mechanical transporters.

Well, although the conclusion is a little bit disappointed, this is the first review that focused on the fish product and three specific zoonotic protozoa, which might be pretty attractive to the readers.

Therefore, as far as I am concerned, this manuscript can be assigned as minor revision. Three concerns and recommendations were listed here.

1) The title is too verbose, especially a review on XX. Spp. etc. is too general, and confusing. It might be direct state is Potential risk of three zoonotic protozoa (XXX, XXX, XXX) transmission from fish consumption, or XXX can be omitted.

2) In the main text, the second part of each section listed several evidence that the specific protozoa was detected from the fish. Are they any regulations from the government to set up its limit in the fish product to be safe consumption? and how to control these protozoa if observed in the fish product.

3) What is the major analytical method to detect these protozoa? It might be good to mention somewhere in the text as well.

Author Response

Valencia, December 15th, 2020

To,

Editor of Foods

 Dear Editor,

I am sending the revised version of the manuscript entitled “Potential risk of zoonotic protozoa transmission from fish consumption: a review on Cryptosporidium spp., Giardia duodenalis and Toxoplasma gondii”, intended for publication in Foods as a review paper.

Please find attached to this cover letter a revised version of the manuscript, thoroughly prepared to accommodate the comments and recommendations made by the Reviewers.    
For your convenience, the detailed list of all the changes, together with the answers to the comments raised by the Reviewers, is provided bellow.

Response to Referee 3 comments:

We sincerely thank the reviewer for the constructive comments and suggestions, which helped us to substantially improve our manuscript. Please find the point-to-point responses (blue) to the comments (black) as listed below

  1. The title is too verbose, especially a review on XX. Spp. etc. is too general, and confusing. It might be direct state is Potential risk of three zoonotic protozoa (XXX, XXX, XXX) transmission from fish consumption, or XXX can be omitted.

The title has been modified according to the suggestion to make it clearer for readers. We have decided to maintain the specific protozoa species in the title; we consider important to remark which ones we have considered, because we are talking about a “new” source of infection (fish) for these protozoa.

  1. In the main text, the second part of each section listed several evidences that the specific protozoa was detected from the fish. Are they any regulations from the government to set up its limit in the fish product to be safe consumption? and how to control these protozoa if observed in the fish product.

As we mentioned at the end of the Introduction, there are no regulations concerning protozoa in any type of food, including fish. Therefore, presence of Cryptosporodium spp., Giardia duodenalis and Toxoplasma gondii in fish is not analysed in the industry. Nonetheless, a correct cooking would eliminate the potential risk, being the problem in raw or undercooked products.

  1. What is the major analytical method to detect these protozoa? It might be good to mention somewhere in the text as well.

We have added some more information at the end of the introduction to explain this question (lines 76-80). Briefly, there no standard methods that can be routinely applied for the detection of these protozoa in food (there is only one exception explained in the new text). On the other hand, and following another referee suggestion, we have added three new tables in subsections 2.1, 3.1 and 4.1 (Tables 4, 7 and 8, for Cryptosporidium spp, Giardia duodenalis and Toxoplasma gondii, respectively). Each one of these three tables summarizes the molecular methods used in the different studies concerning these protozoa in fish (type of sample, molecular technique and target genes are shown for each study). However, this molecular technique refers to research purposes.

Round 2

Reviewer 1 Report

The manuscript has fully revised and addressed all the issues which has significantly improved the quality of the topic for the reader. Therefore, I strongly recommend to publish on Food journal.